# Wind Direction Retrieval Using Support Vector Machine from CYGNSS Sea Surface Data

**Yun Zhang** [1], **Xu Chen** [1], **Wanting Meng** [2], **Jiwei Yin** [1], **Yanling Han** [1], **Zhonghua Hong** [1] **and Shuhu Yang** [1,*]

1   Shanghai Marine Intelligent Information and Navigation Remote Sensing Engineering Technology Research Center, Shanghai Ocean University, Shanghai 201306, China; y-zhang@shou.edu.cn (Y.Z.); m200711435@st.shou.edu.cn (X.C.); m190711309@st.shou.edu.cn (J.Y.); ylhan@shou.edu.cn (Y.H.); zhhong@shou.edu.cn (Z.H.)
2   Shanghai Spaceflight Institute of TT&C and Telecommunication, Shanghai 201109, China; wanting_meng@163.com
*   Correspondence: shyang@shou.edu.cn

**Abstract:** In view of the difficulty of wind direction retrieval in the case of the large space and time span of the global sea surface, a method of sea surface wind direction retrieval using a support vector machine (SVM) is proposed. This paper uses the space-borne global navigation satellite systems reflected signal (GNSS-R) as the remote sensing signal source. Using the Cyclone Global Navigation Satellite System (CYGNSS) satellite data, this paper selects a variety of feature parameters according to the correlation between the features of the sea surface reflection signal and the wind direction, including the Delay Doppler Map (DDM), corresponding to the CYGNSS satellite parameters and geometric feature parameters. The Radial Basis Function (RBF) is selected, and parameter optimization is performed through cross-validation based on the grid search method. Finally, the SVM model of sea surface wind direction retrieval is established. The result shows that this method has a high retrieval classification accuracy using the dataset with wind speed greater than 10 m/s, and the root mean square error (RMSE) of the retrieval result is 26.70°.

**Keywords:** GNSS-R; wind direction; CYGNSS; SVM; DDM



## 1. Introduction

Wind speed and wind direction are important basic climate variables. As a passive and non-contact remote sensing method, the Global Navigation Satellite System Reflectometry (GNSS-R) technique uses the reflected signal of the navigation satellite L-band signal as the remote sensing source. The GNSS-R technique is based upon a bistatic configuration of the transmitter and receiver. The scattering problem involves the Global Positioning System (GPS) signals transmitted from satellites at altitudes of about 20,000 km [1]. The system has outstanding application effects in retrieving various physical parameters, such as sea surface wind speed [2,3], sea ice [4–6], ocean surface altimetry [7–9], ocean oil slick detection [10,11]. Sea surface wind field retrieval is mainly divided into sea surface wind direction retrieval and sea surface wind speed retrieval. Sea surface wind direction retrieval is a difficult point in wind field retrieval. Few studies involve the sensitivity of GNSS-R signals to wind direction [12].

Many studies have focused on finding feature parameters closely related to wind direction from the Delay Doppler Diagram (DDM). Early research was mainly based on the relationship between signal delay waveform, probability density function (PDF) and wind direction. In 2003, the influence of wind direction on the GPS signal delay waveform by analyzing the reflected waveform geometry, scattering area and receiver integration time [13,14] was simulated by Zuffada et al. Given the satellite elevation angle, wind speed and receiver height, Zuffada et al. found that there were significant differences in different wind directions at the trailing edge of the waveform. In 2004, similar results were shown

in the carrier shape of the simulator at the height of 10 km. Further research found that the trailing edge of the waveform of left-hand circular polarization (LHCP) and right-hand circular polarization (RHCP) had differences [15]. When the incident angle increased, this difference would be reduced. In 1994, Hildebrand et al. used the least-squares method to perform wind direction retrieval based on PDF when the aircraft's receiver height is 3–5 km, the highest wind direction retrieval accuracy on the filtered data set and the unfiltered data set is 5° and 40° [16]. On this basis, in 2004, the anisotropy in the PDF of the mean square slope (MSS) of the sea surface was found to correspond to the local near-surface wind direction. The PDF can be calculated from the code correlation waveform of the GPS reflected signal, which showed the wind direction can be obtained by using the scattered signals of two or more GPS satellites [17]. In 2003, the MSS was estimated by fitting the delayed waveform obtained in the flight experiment to the geometric optical model [18]. The part of the wind direction when the satellite azimuth was consistent with the aircraft heading was the same as that of the European Centre for Medium-Range Weather Forecasts (ECMWF) wind direction data. In 2004, Komjathy et al. used different satellite signals collected from airplanes combined with the nonlinear least-squares algorithm to retrieve the wind direction. The result showed that the retrieved wind direction and the QuikSCAT measurements were in 30° error at a wind speed of 5–10 m/s when the airplane had both a stable flight level and a stable flight direction [19]. In 2014, by fitting the measured DDM with the simulated DDM, Chen et al. made the accuracy of the wind direction retrieval reach 30° [20].

Another research work on wind direction retrieval was mainly focused on the surface reflection signal under the general mirror geometry observation configuration. In 2017, the influence of wind direction on the near-specular reflection area of DDM was used in calculating the Delay Doppler Map Average (DDMA) from the perspective of DDMA [21]. The results showed that for pure mirror geometry, DDMA had an effect on wind direction. The relation between DDMA and wind direction is small and the influence of wind speed cannot be ruled out. When the incident angle was 20°–25° and the wind speed was about 9 m/s, the normalized signal-to-noise ratio (SNR) peak difference between the tailwind and the headwind was 1 dB. However, it was currently difficult to retrieve the wind direction through these small changes [22]. However, refs. [20,21] both found differences in the dependence of different parts of the DDM on the wind direction in the process and predicted the possibility of using the part of the DDM far away from the specular reflection point to retrieve the wind direction.

At present, more studies tend to use the relationship between wind direction and the bistatic radar cross-section. In 2014, the sensitivity of the bistatic radar cross-section to the wind direction was evaluated using a small slope approximation model [23]. The model was more accurate when using scattered signals away from the nominal specular reflection direction. In 2016, Park et al. used a Normalized Bistatic Radar Cross Section (NBRCS) to study the effect of wind direction on GNSS-R application sea surface specular scattering [24]. For purely mirrored geometry, the change in NBRCS was too small to perceive the wind direction. For only slightly non-mirrored geometry, a large change in wind direction can be found at a single surface point. An airborne wind direction retrieval model based on NOAA G-IV jet aircraft using the Doppler angle of DDM as the retrieval observation was established in the study [25]. The average accuracy of wind direction retrieval obtained under a fixed model was 20°. However, the model was a non-general model with a small amount of airborne data, and there was also a problem of 180° ambiguity. In 2018, Wang et al. explored the feasibility of using backscattered signals to retrieve wind direction through theoretical simulations, using multi-beam antennas to observe wind direction from at least three different directions to avoid ambiguity. The retrieval accuracy reached 24° at low wind speeds [26]. In 2021, Pascual et al. drew the conclusion that wind speed and SNR had an important influence on the wind direction retrieval model. The sensitivity of the ocean surface bistatic scattering cross-section measured by the Cyclone Global Navigation Satellite System (CYGNSS) to wind direction using the kurtosis of the DDM samples within

a given area was studied. The results show a coefficient of determination ($R^2$) between 0.6 and 0.9 for wind speeds between 4 and 10 m/s [27].

According to the above research results, it is difficult to establish a sea surface wind direction retrieval model, especially in the case of a large space and time span. The solution of wind direction 180° ambiguity is also a key point in wind direction retrieval. Artificial intelligence algorithms, such as machine learning and deep learning, have made it possible to build complex models. Convolutional Neural Networks (CNN) is a typical algorithm in deep learning, and support vector machine (SVM) is a typical algorithm in machine learning [28–30]. Due to the small number of training samples in this paper, the SVM algorithm is chosen. A small number of support vectors determine the final result during SVM calculation, which is insensitive to outliers and has excellent generalization capabilities [31].

After our research about the sea surface wind speed inversion model of the CYGNSS sea surface data based on Machine Learning [32], this paper studies the sea surface wind direction retrieval model of space-borne GNSS-R based on SVM. The data comes from CYGNSS Full DDM data, CYGNSS L1 data and ECMWF reanalysis datasets from 2019 to 2020. The geometric relationship parameters of DDM and CYGNSS satellite parameters are extracted as SVM feature parameters. The grid search method is adopted to optimize and establish a global satellite-borne sea surface wind direction retrieval model and verify the effectiveness of the model. Compared with the abovementioned research, the space-borne GNSS-R sea surface wind direction retrieval has a wider detection range and a larger amount of data. The CYGNSS satellite data selected in this paper has a wide coverage and a large time span. The results show that the SVM method proposed in this paper can effectively retrieve the sea surface wind direction.

## 2. Data Source

### 2.1. CYGNSS

The "CYGNSS" mission launched on 15 December 2016, and placed eight small satellites in low-Earth orbit at an altitude of about 520 km [33]. Each satellite carried the CYGNSS Delay Doppler Mapping Instrument (DDMI) payload, which received GPS signals reflected from the Earth's surface. Therefore, each satellite receiver operated a bistatic radar to observe the forward scattering of a specific GPS transmitter from the Earth's mirror. By correlating the received GPS transmission with the sent locally-generated copy of the GPS C/A code, a basic GNSS-R measurement or DDM, was formed. The "pixel" in DDM represents the power shifted from the Earth's surface with a certain delay and Doppler scattered from the specular reflection point. Each CYGNSS satellite was equipped with two nadir antennas, capable of tracking four simultaneous reflections at any given time. In the range of ≈±38°, the constellation has produced up to 32 DDM measurements every 0.5 s since 2019. Due to the movement of the CYGNSS receiver and GPS transmitter, the CYGNSS measurements are performed on irregularly repeated orbits in time and space; the average and median revisit times are 7 and 3 h. These features have made CYGNSS measurement conducive to achieving frequent storm coverage.

The CYGNSS data file used in this paper includes FULL DDM and L1 data, including time-delay Doppler sample data, a series of metadata of geometric structure and instrument parameters at the time of acquisition and the data format is NetCDF. This paper uses version 2.1 of the CYGNSS L1 data (available at https://podaac.jpl.nasa.gov/dataset/CYGNSS_L1_V2.1). The time period is from 27 February 2019, to 17 November 2020. DDM sample data is extracted from FULL DDM data, and CYGNSS satellite parameters are extracted from L1 data. FULL DDM data and L1 data are based on the index that corresponds to the specular reflection point for data matching.

### 2.2. ECMWF Reanalysis Data Set

ECMWF used its forecasting models and data assimilation system to reanalyze archived observations to create a global data set describing the recent history of the at-

mosphere, land and ocean. The global data assimilation and filtering system used in the ECMWF reanalysis project had continuity. A database that is as complete and comprehensive as possible ensures the consistency of the reanalysis data. The data for reanalysis came from radio detection, aircraft, satellites and ground stations. The comprehensive use of multiple data sources can eliminate possible obstacles or vegetation and other local effects of non-target observations. The data was filtered to ensure data independence and consistency.

ECMWF provides a sea surface wind field data of U and V components at the height of 10 m above the sea surface. ECMWF provides sea surface wind field data with a spatial resolution of $0.25° × 0.25°$ and a time resolution of 1 h. This paper used version ERA5 of the ECMWF data. The dataset matches the wind field data of the closest point in space and time to the CYGNSS specular reflection point. The interpolation method is not used in this paper because the error between the nearest neighbor-matched wind direction and the interpolated wind direction is small. Selecting the nearest neighbor matching method can reduce the amount of calculation. In this paper, the wind speed data provided by ECMWF is used as training data, and wind direction data is used as truth data.

## 3. Geometric Feature Parameter Extraction

According to the GNSS-R bistatic radar equation, wind direction causes a change in the direction of the sea surface slope resulting in DDM asymmetry, but other parameters can also contribute to the shape of DDM [34]. Therefore, it is difficult to extract the wind direction from the DDM directly, and the geometric feature parameter of the DDM, which is more sensitive to wind direction and less sensitive to other factors, is needed. Most research and observations have shown that the DDM expands when the wind speed increases. When the DDM expands, the DDM peak, the center of mass of $DDM_r$ and the center of mass of $DDM_{skirt}$ will shift toward larger delay bins. The difference of their change is more determined by wind direction than by other factors, such as wind speed. $DDM_r$ is to filter out DDM parts with a horseshoe shape whose power value is greater than the specified threshold after normalizing the DDM. $DDM_{skirt}$ is the region whose power is within a certain ratio of the peak value of DDM.

Therefore, two measurement parameters to establish the relationship with the wind direction are proposed. The first parameter is the vector azimuth angle from the peak point of DDM to the center of mass of $DDM_r$, namely angle φ1 in Figures 1–4. The second parameter is defined as the vector azimuth angle from the center of mass of $DDM_r$ to the center of mass of $DDM_{skirt}$, namely angle φ2. The peak point corresponds to the specular reflection point.

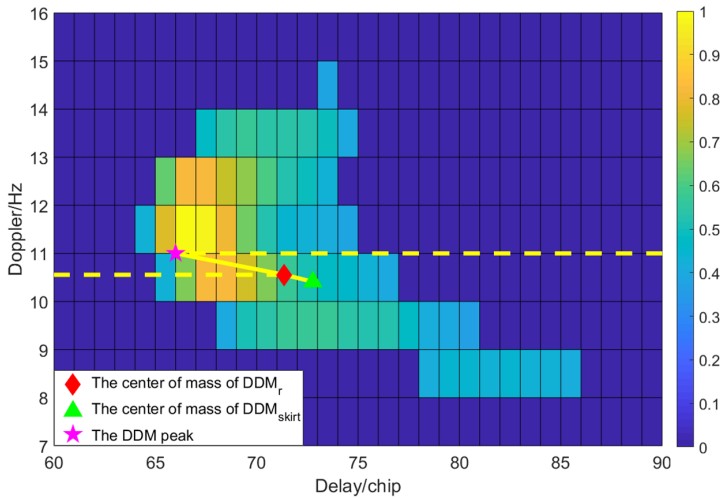

**Figure 1.** Extracting feature parameters of the geometric relationships φ1 and φ2 from the DDM of a 0° wind direction.

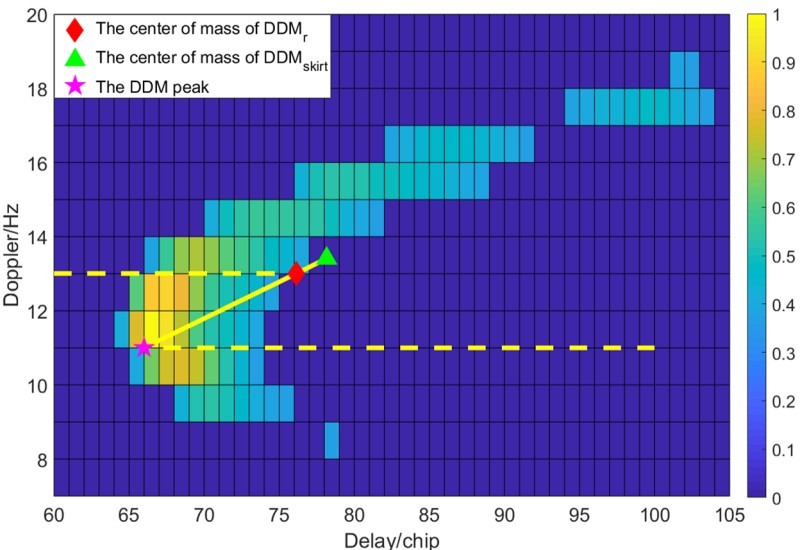

**Figure 2.** Extracting feature parameters of the geometric relationships φ1 and φ2 from the DDM of a 90° wind direction.

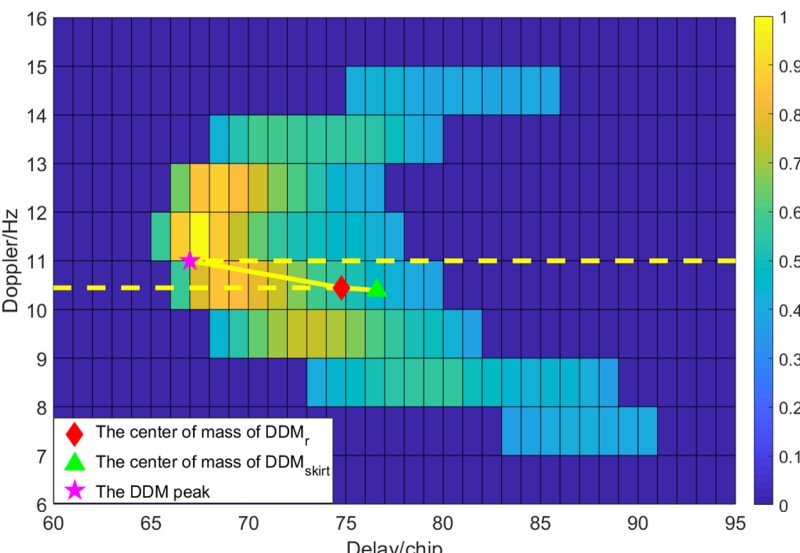

**Figure 3.** Extracting feature parameters of the geometric relationships φ1 and φ2 from the DDM of a 180° wind direction.

When calculating these two indicators, the DDM power is normalized based on the DDM peak value to avoid any calibration problems. The specific steps are as follows:

(a) Normalize DDM based on the peak point.

(b) Filter out the $DDM_r$ part with normalized power value greater than $e^{-1}$ from DDM.

(c) Calculate the coordinates of the DDM peak point ($X_{max}$, $Y_{max}$) and the centroid coordinates ($X_{cm}$, $Y_{cm}$) of $DDM_r$, where $X$ represents the time delay, $Y$ represents the Doppler shift and $T(X, Y)$ represents the corresponding power at the time delay $X$ and the Doppler shift $Y$. The calculations of $X$ and $Y$ are shown in Equations (1)–(3).

$$X_{cm} = T_0^{-1} \sum \sum X \cdot T(X,Y) dX dY \tag{1}$$

$$Y_{cm} = T_0^{-1} \sum \sum Y \cdot T(X,Y) dX dY \tag{2}$$

$$\text{where} \quad T_0 = \sum \sum T(X,Y) dX dY \tag{3}$$

(d)   Filter out the part of $DDM_{skirt}$ whose power value is 30% to 70% of the peak power value from $DDM_r$. This paper adjusts the threshold obtained after the attempt from the statistical analysis in a large number of DDM during the calculation of φ1 and φ2. Then calculate the center of mass of $DDM_{skirt}$ in the same way as above.

(e)   Calculate φ1 and φ2 from the DDM peak point, the center of mass of $DDM_{skirt}$ $(X_{cm}, Y_{cm})$ and the center of mass of $DDM_{skirt}$ $(X_{cmskirt}, Y_{cmskirt})$ to obtain geometric feature parameters.

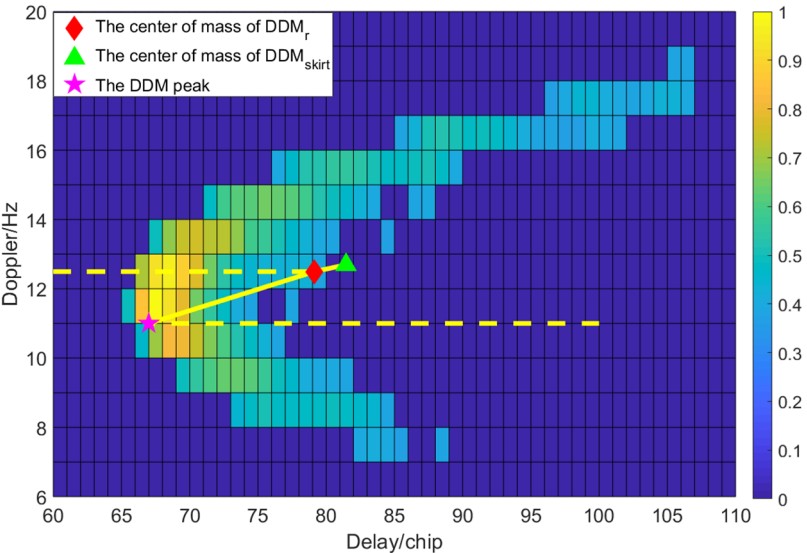

**Figure 4.** Extracting feature parameters of the geometric relationships φ1 and φ2 from the DDM of a 270° wind direction.

Figures 1–4 show that the shape of the DDM with different wind directions is significantly different between Figure 1 (or Figure 3) and Figure 2 (or Figure 4). The part outside the dark blue area is $DDM_r$. The five-pointed star represents the peak point, the diamond represents the center of mass of $DDM_r$ and the triangle represents the center of mass of $DDM_{skirt}$.

The two angle parameters are related to the sea surface wind direction through the statistical analysis of a large amount of data of CYGNSS Full DDM. Figures 1–4 are typical examples representing different wind directions. Therefore, φ1 and φ2 can be used to retrieve the wind direction. However, it also can be seen that wind direction being retrieved by φ1 and φ2 presents a double ambiguity of 180°. In order to avoid the ambiguity of wind direction and improve the accuracy of wind direction retrieval, other parameters related to wind direction are needed.

Figures 1–4 also indicate that the normal angle range of φ1 should be between 0° and 10°, and the normal angle range of φ2 should be between 170° and 180°. Based on a large amount of statistical data, this paper finds that most of the angles of φ1 that can reflect the asymmetry of DDM are between 0° and 10° and for φ2 are between 170° and 180°, which belong to normal angles. In order to improve the robustness of the model, this paper slightly expands the range of normal angles. Therefore, this paper defines φ1 that is greater than 15° and φ2 that is less than 165°, as thresholds of abnormal angles. The abnormal angle includes most angles that cannot correctly reflect the asymmetry of DDM. In most cases of abnormal angles, φ1 and φ2 cannot be used to establish a connection with wind direction. However, some data samples with abnormal angles can still be used, and this paper does not simply delete data samples with abnormal angles according to numerical values. Specific data preprocessing conditions will be discussed later.

## 4. Wind Direction Retrieval Based on Multi-Dimensional Feature Vector

### 4.1. GNSS-R Feature Parameters

The relevant power waveform of the GNSS-reflected signal is not only related to the sea surface wind speed but also related to its corresponding GNSS-R geometric relationship. Based on this, this paper extracted nine dimensional feature parameters of GNSS-R events. They contain not only the CYGNSS satellite parameters but also the geometric relationship feature parameters of the DDM waveform. In order to establish a more universal wind direction retrieval model, it is necessary to establish the mapping relationship between the feature parameters and the sea surface wind direction (*WD*). This paper established two mapping relationships as follows:

$$WD_1 = f_1(WS, \theta, \mu, \varphi1, \varphi2) \tag{4}$$

$$WD_2 = f_2(WS, \theta, \mu, \varphi1, \varphi2, LES, NBRCS, SNR, RCG) \tag{5}$$

where the corresponding feature parameters are wind speed (*WS*), receiver elevation ($\theta$) and azimuth angle ($\mu$), $\varphi1$, $\varphi2$, leading-edge slope (*LES*), *NBRCS*, *SNR* and Range Corrected Gain (*RCG*). From Equations (4) and (5), $f_2$ uses four more feature parameters than $f_1$. The ws, ele and azi are important parameters for retrieving the wind direction from DDM. Some good results can be obtained by using Equation (4). However, these parameters cannot fully reflect the influence of sea surface roughness on DDM, so *LES*, *NBRCS*, *SNR* and *RCG* are introduced in Equation (5). Limited by the influence of multi-dimensional parameters, the analytic function of formulas is very complicated, but machine learning theory provides convenience for training multi-parameter nonlinear regression models. Therefore, in order to fully consider the relationship between DDM geometric relationship feature parameters, wind speed, CYGNSS satellite parameters and wind direction, this paper will introduce the SVM algorithm to establish a mapping relationship model between the above parameters and perform sea surface wind direction retrieval.

### 4.2. SVM

SVM is a widely used supervised machine learning algorithm in statistical learning. The two basic theories of statistical learning theory, "VC Dimension Theory" and "Structural Risk Minimization Theory" are the source of its basic ideas and have obvious advantages in solving nonlinear classification problems and high-dimensional feature space classification problems. The key of the SVM algorithm is to find the best feature space separation hyperplane. Specifically, $K(x, z)$ is a function or positive definite kernel, which means that there is a mapping $\varphi(*)$ from the input space to the feature space. For any input space, there is:

$$K(x, z) = \varphi(x) \bullet \varphi(z) \tag{6}$$

In the dual problem of linear SVM, the kernel function $K(x, z)$ is used to replace the inner product, and the nonlinear SVM obtained by the solution is as follows:

$$f(x) = sign(\sum_{i=1}^{N} \alpha_i^* y_i K(x, x_i) + b^*) \tag{7}$$

Based on the above discussion, the nonlinear SVM learning algorithm can be obtained as follows:

Input: training dataset $T = \{(x_1, y_1), (x_2, y_2), ..., (x_n, y_n)\}$

Output: Separate hyperplane and classification decision function

The input parameters of the SVM model in this paper are shown in Equations (4) and (5). *X* represents the feature parameter of five or nine dimensions, and *Y* represents the label of wind direction interval in *T*.

### 4.3. Gaussian Kernel Function

The essence of the Gaussian kernel is to measure the "similarity" between the sample and the sample. In a space that describes the "similarity", the samples of the same kind can be gathered together better and then linearly separable. The Gaussian kernel function has the advantages that it can be mapped to infinite dimensions, the decision boundary is more diverse and there is only one parameter, which is easier to choose compared to the polynomial kernel. Gaussian kernel function definition method:

$$K(x, y) = e^{-\gamma \|x - y\|^2} \tag{8}$$

where $x$ and $y$ are samples or vectors; $\gamma$ is the only hyperparameter of the Gaussian kernel function that represents the norm of the vector, which can be understood as the modulus of the vector.

In this paper, the SVM model uses Radial Basis Function (RBF) with good generality. In order to improve the accuracy and generalization ability of the retrieval model, it is the key to optimizing the kernel function parameter $\Gamma$ and the penalty factor $C$. $\Gamma$ affects the smoothness of the interface. $C$ affects the model's tolerance for data sample misdivision. Therefore, the grid search cross-validation is used to optimize the parameters, and a large search range of $\Gamma$ and $C$ is determined in advance, and then all $\Gamma$ and $C$ on the search grid are traversed for training, and the classification accuracy of the cross-validation under each set of parameters is obtained. Finally, this paper selects a set of parameters with the highest accuracy rate as the training parameter result, which is retrained to build the model. The process is shown in Figure 5.

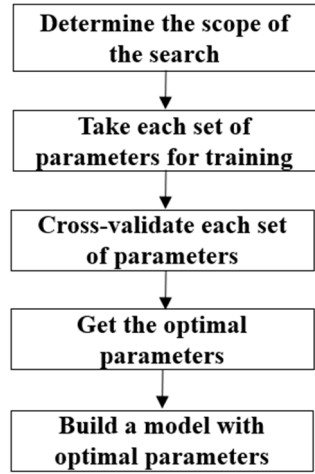

**Figure 5.** Cross-validation based on grid search method.

### 4.4. SVM Sea Surface Wind Direction Retrieval Process

This section proposes an SVM wind direction retrieval algorithm based on multi-dimensional feature parameters. The DDM sample data comes from the Full DDM data of the CYGNSS satellite. The receiver elevation and azimuth angles, LES, RCG, NBRCS and SNR, come from the CYGNSS L1 data of the specular reflection point corresponding to the Full DDM data. The true data of wind speed and wind direction come from the ECMWF fusion wind. All data is matched and calculated in time and space. In wind direction retrieval, the geometric feature parameters, CYGNSS satellite parameters and wind speed data form a training dataset, which is used as the input of SVM. In order to improve the efficiency of calculation and reduce the complexity of the model, the wind direction from 0° to 360° is divided into 72 intervals with a step length of 5°. The wind direction of 0° is the wind direction of true north. True wind direction provided by ECMWF is converted into every interval, and the label of the interval was used to train the wind direction model. The basic retrieval process is shown in Figure 6.

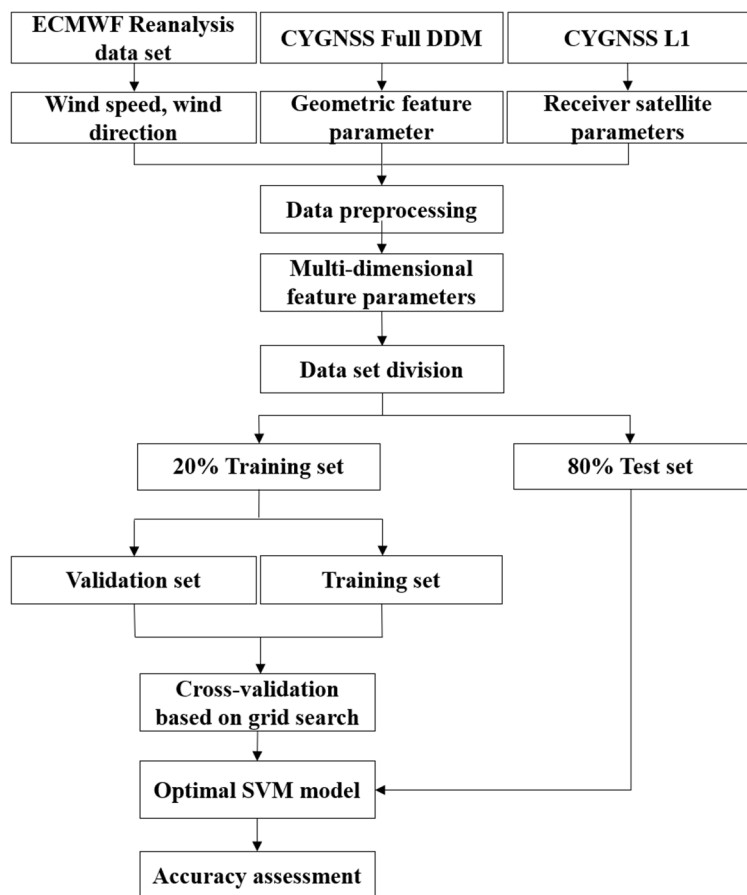

**Figure 6.** GNSS-R wind direction retrieval flow chart based on SVM.

## 5. Data Process and Results

### 5.1. Data Preprocessing

The sample data are from 27 February 2019, to 17 November 2020, in this paper. The data set information (Table 1) is as follows:

**Table 1.** Data set information.

| Data Set | Type | Time Coverage | Temporal Resolution |
|---|---|---|---|
| CYGNSS | L1 | 27 February 2019–17 November 2020 | 0.5 s |
| CYGNSS | Full DDM | 27 February 2019–17 November 2020 | 0.5 s |
| ECMWF | Reanalysis dataset | 27 February 2019–17 November 2020 | 1 h |

The data collection quality of the dataset is basically controlled and preprocessed. The standards of data preprocessing are as follows:

(1) Through the quality control (QC) flag in the L1 data of CYGNSS, this paper selects data samples with good quality.

(2) The power of DDM with low wind speed is mainly concentrated near the specular reflection point, and it is difficult to get the normal angle. Data samples with a wind speed above 5 m/s are selected in this paper, and data are processed in different wind speed ranges above 5, 8, 10, 12 and 15 m/s.

(3) When the SNR is too low, the original shape of DDM is "submerged" in the noise, which affects the data quality of geometric relationship feature parameters ($\varphi 1$ and $\varphi 2$). The abnormal angles ($\varphi 1$ and $\varphi 2$) are defined in Section 3. Figure 7 shows that there is a much larger amount of data samples with an abnormal angle when the SNR is lower than 1.3, so the data sample with an SNR higher than 1.3 is selected here.

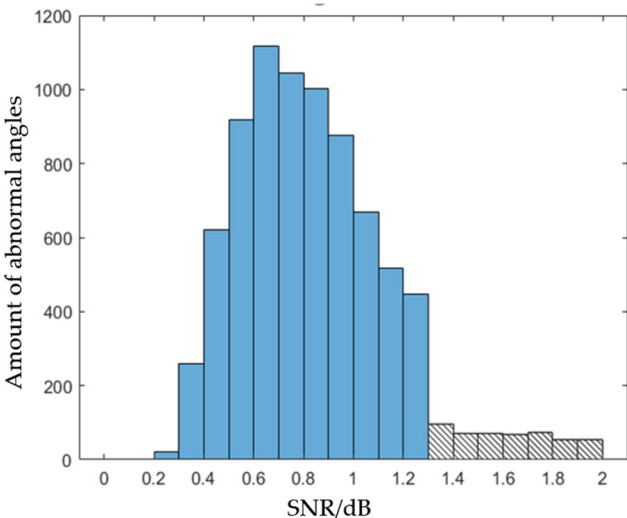

**Figure 7.** The number distribution of abnormal angles in different SNR retrievals.

(4) Wind direction will be affected by the land airflow, resulting in the unusual shape of DDM data, and sometimes even the extreme situation that the peak point is located at the end of the horseshoe shape. Therefore, the sample data with an offshore distance of more than 25km is selected here.

The distribution of the final filtered data on different data sets is shown in Table 2, the quantity distribution of the data in different months before and after the screening is shown in Figure 8, and the distribution of the filtered data in different wind directions is shown in Figure 9. It can be seen from Figure 8 that the data samples are mainly concentrated in July to November, and Figure 9 shows that wind direction is mainly concentrated in the interval of 1–36 (0°–180°).

**Table 2.** Data quantity of different wind speed ranges.

| Wind Speed | ≥5 m/s | ≥8 m/s | ≥10 m/s | ≥12 m/s | ≥15 m/s |
|---|---|---|---|---|---|
| Amount | 127,786 | 57,593 | 29,788 | 16,552 | 6498 |

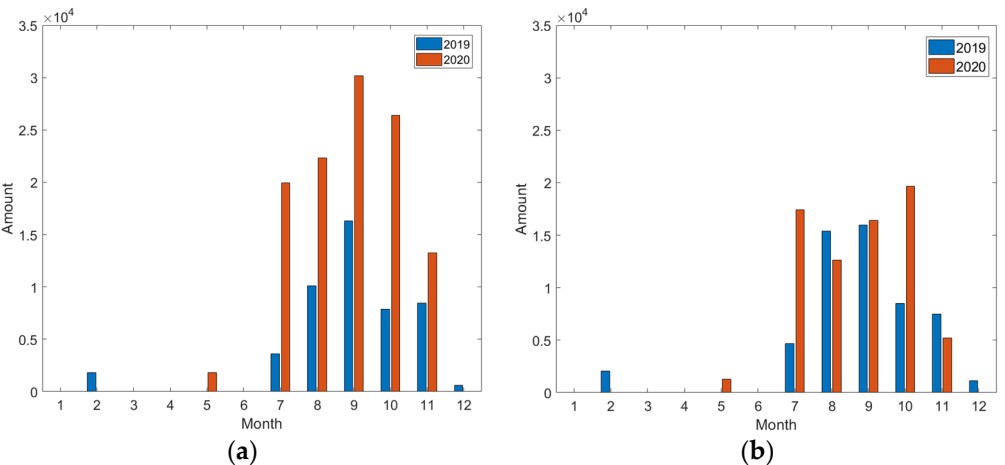

**Figure 8.** Quantitative distribution of Full DDM data in different months before (**a**) and after data preprocessing (**b**).

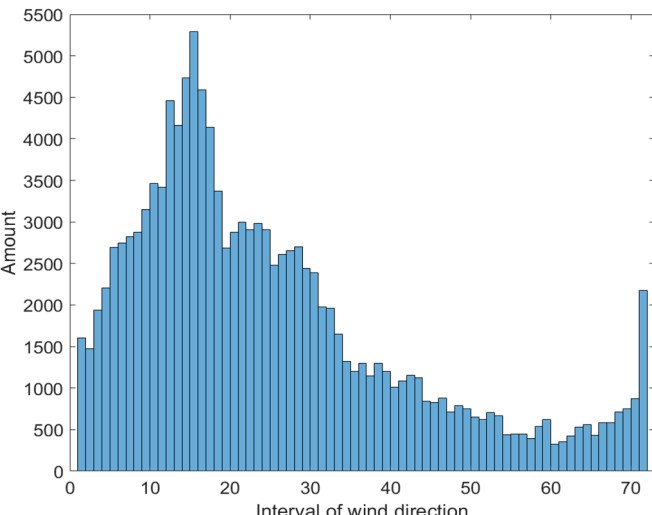

**Figure 9.** Quantitative distribution of filtered Full DDM data in different wind direction intervals.

### 5.2. Accuracy Assessment

According to the flow chart in Figure 6, the SVM sea surface wind direction retrieval model is established. The data set imported here is from CYGNSS satellite and ECMWF reanalysis datasets. The dataset is randomly divided into 80% for the test set and 20% for the training set, and then 20% of the training set is divided for grid cross-validation. The cross-validation grid search method is carried out five times. After obtaining the model, the accuracy results are obtained in the test set, and the root mean square error (RMSE) is calculated as follows:

$$\text{RMSE} = \sqrt{\frac{\sum_{i=1}^{n} diff_i^2}{n}} \tag{9}$$

$$where \quad diff = \left| WD_{predict} - WD_{true} \right| \tag{10}$$

In Equation (9), $n$ is the number of samples. $WD_{true}$ is the true value of the wind direction from the ECMWF reanalysis data set, and $WD_{predict}$ is the middle value of the predicted wind direction interval (the middle value of label 1 interval is 2.5°, and that of label 2 interval is 7.5°).

### 5.3. Retrieval Results of Wind Direction

#### 5.3.1. Grid Search Results

Based on the best parameters obtained by grid search cross-validation, the SVM model is established to evaluate the accuracy of 80% of test sets randomly selected from the dataset. Figure 10 is the result of a grid search for datasets with a wind speed range of more than 8 m/s with nine dimensional feature parameters. It can be seen from Figure 10a that the optimal parameter combination is near the range of $C = 10$ and $\Gamma = 10^{-1.5}$, and the optimal parameter can be obtained by further narrowing the range. For datasets with different wind speed ranges, the accuracy evaluation results are shown in Table 3.

#### 5.3.2. Results Comparison between Two Different Mapping Relationships

Table 3 shows the optimal parameters results (after Grid searching) and RMSEs of SVM retrieval results of different wind speed datasets ($\geq$5, $\geq$8, $\geq$10, $\geq$12 and $\geq$15 m/s) with two mapping relationships, $f_1$ and $f_2$.

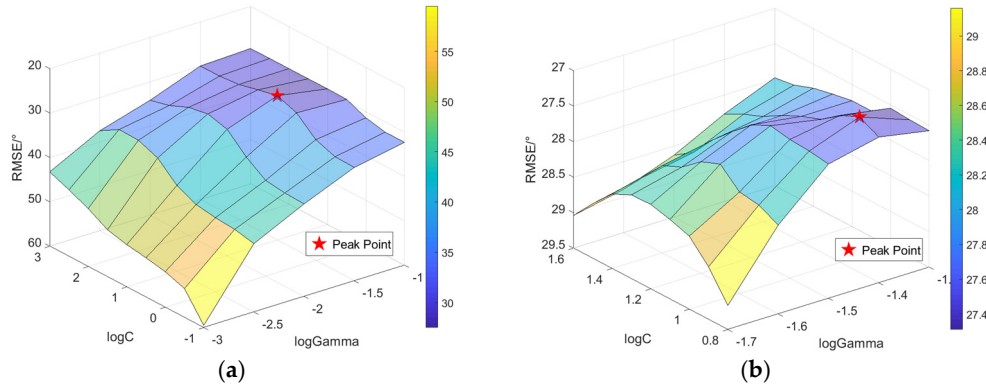

**Figure 10.** Grid search results of datasets with wind speed greater than 8 m/s with nine dimensional feature parameters. (**a**) is rough optimization and (**b**) is fine optimization.

**Table 3.** SVM results of different wind speed datasets with two mapping relationships.

| Mapping Relationship | Wind Speed | Optimal Parameters | RMSE (°) |
|:---:|:---:|:---:|:---:|
| **5 dimensional feature parameters** | | | |
| $f_1$ | $\geq 5$ m/s | $(\Gamma = 10^{-1.2},\ C = 10^{1.7})$ | 49.51 |
| $f_1$ | $\geq 8$ m/s | $(\Gamma = 10^{-1.3},\ C = 10^{1.9})$ | 47.82 |
| $f_1$ | $\geq 10$ m/s | $(\Gamma = 10^{-1.1},\ C = 10^{1.9})$ | 44.12 |
| $f_1$ | $\geq 12$ m/s | $(\Gamma = 10^{-0.9},\ C = 10^{2.1})$ | 41.11 |
| $f_1$ | $\geq 15$ m/s | $(\Gamma = 10^{-1},\ C = 10^{2})$ | 39.51 |
| **9 dimensional feature parameters** | | | |
| $f_2$ | $\geq 5$ m/s | $(\Gamma = 10^{-1.6},\ C = 10^{1.4})$ | 30.03 |
| $f_2$ | $\geq 8$ m/s | $(\Gamma = 10^{-1.4},\ C = 10^{0.9})$ | 27.31 |
| $f_2$ | $\geq 10$ m/s | $(\Gamma = 10^{-1.3},\ C = 10)$ | 26.70 |
| $f_2$ | $\geq 12$ m/s | $(\Gamma = 10^{-1.4},\ C = 10^{1.2})$ | 28.27 |
| $f_2$ | $\geq 15$ m/s | $(\Gamma = 10^{-1.3},\ C = 10^{1.2})$ | 26.78 |

## 6. Results Analysis

### 6.1. Analysis of Data Set Results for Different Wind Speeds

According to Table 3, SVM models with different wind speed ranges have different optimal parameters, and the wind direction retrieval result with a wind speed greater than 10 m/s is best. In the case of a wind speed of 5–10 m/s of $f_2$, with the increase of wind speed, φ1 and φ2 can better reflect the geometric features of DDM asymmetry, which is related to the wind direction, so the RMSE gradually decreases. In the case of a wind speed greater than 10 m/s, the RMSE reaches its lowest. When the wind speed is greater than 12 m/s, the RMSE increases. The main reason is that the data quality decreases with the increase of wind speed, which is reflected in the decrease of SNR and the increase in the number of abnormal angles. When the wind speed range is more than 15 m/s, the RMSE decreases to 26.78°, which is mainly due to the reduction of the amount of data and the concentration of high wind speed data in time and space, resulting in the improvement of classification accuracy of the SVM sea surface wind direction retrieval model.

Compared with $f_2$, the overall classification accuracy of $f_1$ decreased significantly. It shows that the introduction of *LES*, *NBRCS*, *SNR* and *RCG* is conducive to improving the accuracy of wind direction retrieval. These parameters more accurately reflect the variation of sea surface roughness under different wind directions and improve the classification performance of SVM.

According to Table 3, the SVM wind direction retrieval result with a wind speed greater than 10 m/s using mapping-relationship $f_2$ is the best. Figure 11 shows that the SVM classification confusion matrix of the data set with wind direction retrieval result. Most of the data samples are classified into the correct wind direction interval or the range

adjacent to the correct wind direction interval, and the overall classification accuracy is high. It indicates that the SVM sea surface wind direction retrieval model established in this paper has good classification results. However, the RMSE on all datasets is more than 20°, so the results need to be further analyzed.

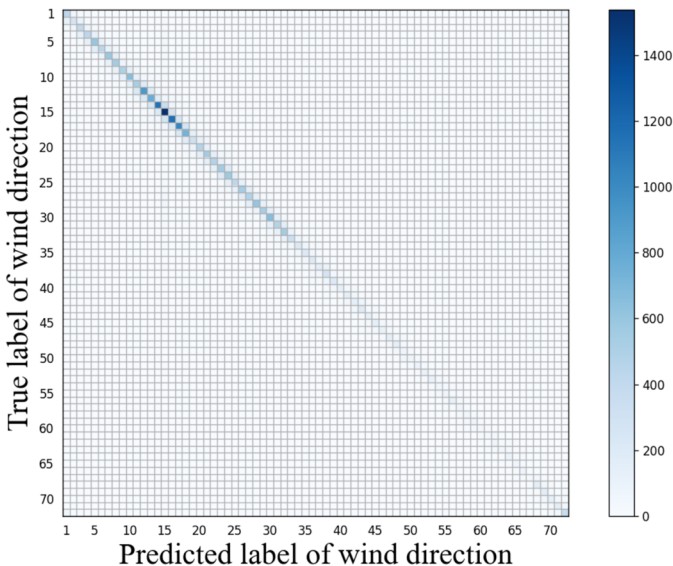

**Figure 11.** Confusion matrix dividing the distribution of SVM classification results into 72 retrievals with wind speeds greater than 10 m/s.

Figure 12 shows the distribution of different angles on the test set. According to Equation (10), the difference can be calculated. The prediction error of most samples is within 5–10°, and only a small part of the data error is within 20–180°. It shows that the SVM model can effectively solve the problem of 180° wind direction ambiguity. However, even if the number of samples with an error of more than 90° accounts for a small proportion of the test set, they still have a considerable impact on RMSE. The biggest penalty for RMSE is to predict a data sample as the opposite wind direction. This is the main reason why the overall classification accuracy of this paper is higher, but the RMSE rises to more than 26°.

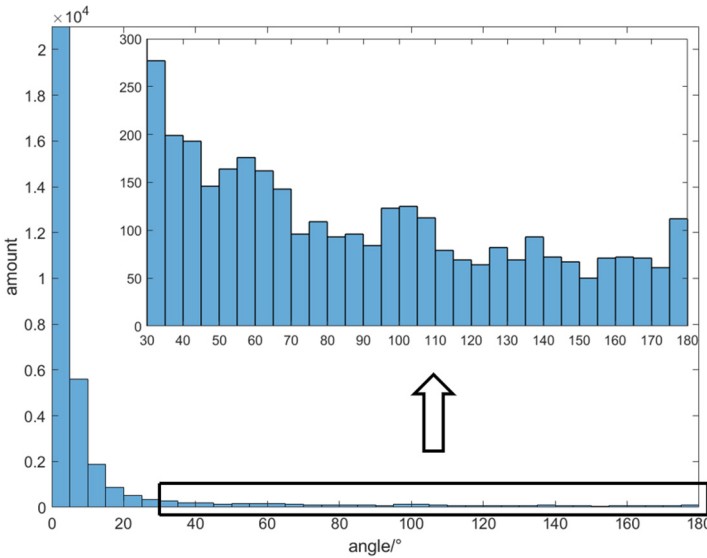

**Figure 12.** Differences in the test set according to sample sequence with a wind speed greater than 10 m/s.

### 6.2. Further Analysis of RMSE Variation under Different Wind Speeds

In the previous section, this paper analyzes the changes of RMSE results under different wind speeds using nine dimensional parameters. Although the actual reason should be as mentioned above, the impact of changes in the amount of data cannot be ignored. With the increase of wind speed, the amount of data is decreasing. It must have a significant impact on the training of the SVM model. Therefore, it is necessary to further analyze the results under the condition of controlling the amount of data.

First, the increase in RMSE for wind speed $\geq$ 12 m/s compared to wind speed $\geq$ 10 m/s should be analyzed. To avoid the impact of data set changes, the under-sampling method was adopted to control the number of data samples. In this paper, 16,552 data samples are randomly selected from the data set in the wind speed between 9 and 12 m/s for the establishment of the SVM model.

From Table 4, the RMSE of the SVM wind direction retrieval model established under the wind speed between 9 and 12 m/s is lower than that of the wind speed greater than 12 m/s. Figures 13 and 14 show the number distribution of SNR in different wind speed ranges. The average SNR in Figure 14 is 3.34. The average SNR in Figure 13 is lower than that in Figure 14, which is 2.78. It indicates that the increase of RMSE wind speed greater than 12 m/s is indeed due to the reduction of data quality compared with wind speed greater than 10 m/s. However, it is also found that reducing the amount of data does have a bad impact on the accuracy. Even in similar wind speed ranges, the reduction of data directly leads to the rise of RMSE.

**Table 4.** Results of wind speed $\geq$ 12 m/s and 9–12 m/s after controlling the amount of data.

| Wind Speed | Optimal Parameters | Amount of Data | RMSE (°) |
|---|---|---|---|
| $\geq$12 m/s | ($\Gamma = 10^{-1.4}$, $C = 10^{1.2}$) | 16,552 | 28.27 |
| 9–12 m/s | ($\Gamma = 10^{-1.6}$, $C = 10^{1.3}$) | 16,552 | 27.15 |

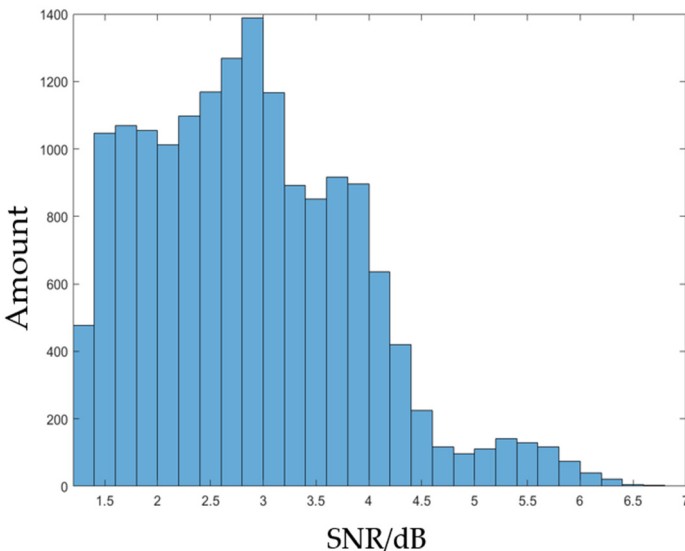

**Figure 13.** The number distribution of SNR with wind speeds greater than 12 m/s.

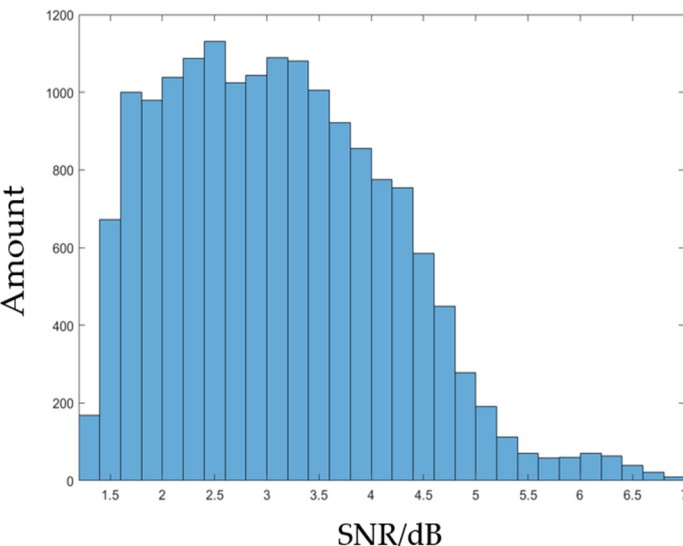

**Figure 14.** The number distribution of SNR with wind speeds between 9 and 12 m/s.

Second, this paper analyzes the decrease in RMSE for wind speed greater than 15 m/s. There are a few data samples with wind speed greater than 15 m/s, and the data quality decreases. The high concentration of data samples in time and space should be the main reason for the decline of RMSE, but the impact of the reduction of data samples also needs to be studied. In this paper, 6498 data samples are randomly selected from the data set in the wind speed range of 12 to 15 m/s for the establishment of the SVM model.

From Table 5, the RMSE at wind speeds of 12 to 15 m/s is higher than that at wind speeds greater than 15 m/s. It shows that under normal circumstances, the reduction of data and the reduction of data quality will lead to the increase of RMSE. However, the high wind speed data samples collected by the CYGNSS satellite are highly concentrated in time and space, which makes the RMSE lower. The SVM model established with this data set is not suitable for the retrieval of all wind directions. The wind direction information contained in the training data set is not complete.

**Table 5.** Results of wind speed $\geq$ 15 m/s and 12–15 m/s after controlling the amount of data.

| Wind Speed | Optimal Parameters | Amount of Data | RMSE(°) |
|---|---|---|---|
| $\geq$15 m/s | $(\Gamma = 10^{-1.3}, C = 10^{1.2})$ | 6498 | 26.78 |
| 12–15 m/s | $(\Gamma = 10^{-1.7}, C = 10^{1.1})$ | 6498 | 32.69 |

This paper established two different mapping relationships between feature parameters and $WD$ ($f_1$ and $f_2$). The results using $f_2$ show that it can effectively solve the problem of wind direction ambiguity. In addition, in order to get $\varphi1$ and $\varphi2$, which can reflect the geometric features of DDM, it is necessary to preprocess the dataset. Firstly, the QC of the CYGNSS L1 data product is used for data preprocessing. Secondly, by analyzing a large number of abnormal angles' data samples, it is found that wind speed and SNR have a greater impact on $\varphi1$ and $\varphi2$. Finally, the selected specular reflection point should be far away from the land. After data preprocessing, the wind direction retrieval based on SVM using nine dimensional feature parameters can accurately retrieve wind direction with a certain condition of wind speed and SNR.

In order to further reduce the RMSE of sea surface wind direction retrieved by the SVM model, more and higher quality DDM data is needed. More can make the model contain all wind direction information, and higher quality DDM can fully reflect the asymmetry of DDM through $\varphi1$ and $\varphi2$. They can improve the accuracy of wind direction retrieval of the model. In addition, this paper retrieves the wind direction based on the asymmetry of DDM. When the wind speed is lower than 5 m/s, the asymmetry of DDM is not obvious.

In the case of low wind speeds and signal interference, the result of wind direction retrieval will become worse. However, based on a large amount of high-quality data, the model is also robust to wind direction retrieval under low wind speeds. Finally, there is one problem that cannot be solved. The sea surface wind direction close to the land cannot be retrieved accurately. Data samples with specular points close to land need to be deleted.

## 7. Conclusions

In this paper, a GNSS-R sea surface wind direction retrieval method based on the SVM model is proposed in the case of a large space and time span. The data are from CYGNSS Full DDM, CYGNSS L1 data and ECMWF reanalysis datasets. By extracting the geometric relationship features $\varphi_1$ and $\varphi_2$ of DDM, the wind direction can be reflected more accurately, which can be used as the important feature parameters of wind direction retrieval. Together with other feature parameters related to wind direction, the input feature parameters of the dataset are composed for the solution of wind direction ambiguity. The wind direction is divided into 72 retrievals in $5°$ steps. Wind speed and SNR have an important influence on the retrieval of sea surface wind direction, especially on the geometric feature parameters of DDM ($\varphi_1$ and $\varphi_2$). Therefore, in order to improve the retrieval accuracy of wind direction, the data of wind speed and SNR are screened and processed.

The retrieval results in different wind speed ranges are evaluated. In order to reflect the impact of sea surface roughness on DDM more accurately, this paper sets up two different mapping relationships, which contain five dimensional feature parameters and nine dimensional feature parameters. Finally, their experimental results are compared. In the case of using nine dimensional feature parameters, when the wind speed is higher than 5 m/s, the RMSE of wind direction retrieved by the SVM model is $30.03°$. When the wind speed is higher than 8 m/s, the RMSE is $27.31°$. In the dataset with wind speed higher than 10 m/s, the RMSE is $26.70°$, which is the best of all. When using five dimensional feature parameters, the overall RMSE in different wind speed ranges increases by more than $10°$, which shows that the introduction of LES, NBRCS, SNR and RCG can effectively improve the accuracy of SVM wind direction classification. RMSEs retrieved under different wind speeds are different. This paper discusses the reasons for the change of RMSE. In order to eliminate the influence of data volume, the down-sampling method is used to control the number of samples. The increase of RMSE from a wind speed greater than 10 m/s to a wind speed greater than 12 m/s is due to the decline of data quality. The decrease of RMSE from wind speed greater than 12 m/s to wind speed greater than 15 m/s is due to the high concentration of data samples in time and space. The results show that a sea surface wind direction retrieval model based on SVM can effectively retrieve the sea surface wind direction and solve the problem of wind direction ambiguity. The spatial-temporal discontinuity of full DDM data, the relatively small amount of filtered data and the error of wind speed products will affect the results of sea surface wind direction retrieval. With the increase of CYGNSS data products, the accuracy of the wind direction retrieval method based on SVM should be further improved.

**Author Contributions:** J.Y. conceptualization; Y.Z. and Y.H. conceived and designed the framework of the study. X.C. completed the data collection and processing. Y.Z. and X.C. completed the algorithm design and the data analysis and were the lead authors of the manuscript, with contributions from X.C., W.M., Z.H. and S.Y. All authors have read and agreed to the published version of the manuscript.

**Funding:** This work was supported by the National Natural Science Foundation of China (Grant No. 41871325, 42176175) and the National Key R&D Program of China (Project No. 2019YFD0900805).

**Institutional Review Board Statement:** Not applicable.

**Informed Consent Statement:** Not applicable.

**Data Availability Statement:** The raw/processed data required to reproduce these findings cannot be shared at this time as the data also forms part of an ongoing study.

**Acknowledgments:** Thanks to NASA for the CYGNSSS public data; the European Centre for Medium-Range Weather Forecasts (ECMWF) for the reanalysis dataset; and the Department of Water Resources of Zhejiang Province for the typhoon track data. We would also like to thank Yang Dongkai and Wang Feng of Beijing University of Aeronautics and Astronautics and Li Weiqiang of CSIC-IEEC for their suggestions on the GNSS-R satellite data analysis. We would like to thank Zhou Bo and Qin Jin from Shanghai Institute of Aerospace Electronics for their suggestions on the receiver of reflected signals.

**Conflicts of Interest:** The authors declare no conflict of interest.

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
