# Peer review of "Wind Direction Retrieval Using Support Vector Machine from CYGNSS Sea Surface Data"

_remotesensing, doi:10.3390/rs13214451_

Round 1

Reviewer 1 Report

This paper presents interesting research on wind direction retrieval using support vector machine. Please consider addressing my below major concerns.

General: Please make sure the labels of Table and Figure are consistent in the entire paper. You have “Table II”, ‘table 3’ in the manuscript. Please add space between number and its unit.

Page 2

What do you mean for ‘scatter meter measurement’?

‘Chen raised the accuracy of the wind direction retrieval to 30 degrees.’ Looks like it’s about the same with previous results.

Page 3

‘has made it possible’ should be ‘have’

“The constellation produces up to 32 DDM measurements per second”. This is not true because CYGNSS changed its sampling frequency to 2 Hz about 2 years ago.

Page 4

‘The dataset matches the wind speed data of the closest point in space and time to the CYGNSS data point’. The typical way is to do interpolation to the ECMWF data in both spatial and time domain. The resolution is too coarse for direct matchup with CYGNSS data. Could you please do the interpolation or at least assess the errors in this process?

How do you pick 30% and 70% for the thresholds?

Page 5

Add space after ‘)’

Page 6

You gave the definition of these points in Figure 1-4 and then say that the two angles can be used to retrieve the wind directions. I don’t feel that you can draw a conclusion from four examples without any further discussion.

Also, please add more content about how you define the abnormal angles.

Page 7 and 8

The two sub-sections of SVM and GNSS-Feature parameter are not closely related. The authors do not know the exact parameters that are input to SVM models. Please add more details.

Page 9

Figure 6: The 80% test set needs to be input to SVM model and then the results will be assessed. Please revise the flow chart.

Table 1: The CYGNSS temporal resolution became 1/2 second about 2 years ago. Please double check and correct this.

Grammar check: ‘The power of DDM with low wind speed is mainly concentrated near the specular reflection point, it is difficult to get the normal angle.’

Page 10

Figure 7: Does this mean that only a very small amount of data can be used to retrieve wind direction? Please give the percentage in the text.

Page 11

Equation (9) and (10) can be combined if you use both superscript and subscript.

Author Response

Dear reviewer,

Thank you very much for your advice. According to your suggestion, the paper has been revised.Please see the attachment. Please see the attachment.

Kind regards

Reviewer 2 Report

This paper presents a machine learning algorithm for wind direction retrieval using space-borne GNSS-R data, namely the CYGNSS data. The authors compare the ECMWF true wind direction value with two predicted values by evaluating the RMSE. The first predicted wind direction value WD1 is a 5-dimensional feature parameter value that doesn’t take into account the leading edge slope, the NBRCS, SNR, and the RCG. The paper then introduces WD2, which is a 9-dimensional feature parameter value that takes the preceding information into account, in addition to the elevation, azimuth, wind speed, and abnormal angles information that is shared between both predictors (WD1 and WD2). The paper then proceeds to introduce a supervised machine learning algorithm defining its training and test sets. This algorithm is tested using CYGNSS real data. The results show that WD2 indeed provides more accurate wind speed information than WD1. The paper comments that the addition of the aforementioned parameters is the reason for such improvement. The result shows that the proposed method has a high retrieval classification accuracy specifically when the wind speed is between 10m/s and 12m/s where the RMSE is the lowest at 26.70°. The paper is well organized and the overall aim of the paper is clear and can constitute a great contribution to space-borne GNSS-R studies. The introduction provides a fair amount of literature reviews. The methods and algorithms introduced are well defined. The results are demonstrated by tables and figures along with the necessary analysis. Saying that, I would like to raise some points below: 1. The paper states that the proposed 9 dimensional feature parameters method has a high retrieval classification accuracy with wind speed greater than 10m/s. However, is that what the results in practice show? From the results, we can indeed notice that the 9-dimensional feature parameters method is better than the 5-dimensional method in almost any wind speed especially when the wind speed is between 10m/s and 12m/s where the RMSE is the best. However, we then observe a significant increase in the RMSE when the wind speed is between 12m/s and 10m/s, and a decrease again when the wind speed is beyond 15m/s regarding the proposed method. These variations are also accompanied by a significant decrease in the amount of data used to process the results. In my opinion, for a clear assessment of the performance of the proposed method, the amount of data shouldn’t be varying that much and I will demonstrate why: • First, the increase in RMSE for wind speed >12m/s compared to wind speed >10m/s: The paper states that the main reason for the increase is that the data quality decreases with the increase of wind speed, which is reflected in the decrease of SNR and the increase of the number of abnormal angles. While that might be the actual reason of the increase, it is worth noting that there is also a decrease of about 13,500 samples between both data sets, which surely will affect the result. My first question is that, is this increase due to the aforementioned reason or due to the decrease in the amount of data used? What is also surprising is that the effect is not consistent with what we observe with wind speed >15m/s. • Second, the decrease in RMSE for wind speed >15m/s: The paper states that the decrease is mainly due to the reduction of the amount of data and the concentration of high wind speed data in time and space, resulting in the improvement of classification accuracy of SVM sea surface wind direction retrieval model. The second question is that why has the decrease in the amount of data led to a decrease in RMSE in this case and to an increase in the previous case? And why we can’t say that there is also a decrease in SNR and an increase in abnormal angles in this case just like the previous one? • Third, the amount of data is significantly decreasing, as the wind speed is getting higher even for wind speeds lower than 10m /s. This also makes me question whether this decrease had a huge impact on the results in equal to the introduction of the additional parameters. This can also be seen in WD1 data where the RMSE is always decreasing. 2. Based on the preceding point, I suggest that for a clear analysis of the performance of the proposed method by itself, an additional study concerning varying wind speeds should be conducted with a consistent amount of data to better analyze the results. 3. In my opinion, the aim of Figure 11 is not clear and should be better explained. 4. I am aware that the proposed method has indeed decreased the RMSE compared to the 5-dimensional feature parameter method. However, the RMSE is still considerably high. The authors state in the last section the reason of such high RMSE, but I would like that the paper also comments on what are, in the authors’ opinions, the future works and perspectives to further decrease this value (maybe in a discussion section). 5. The paper should use scientific notations for representing the variables. For example use Γ instead of Gamma, θ instead of ele… 6. Minor typos should be corrected. Please see the below examples: Abstract: corresponding instead of corresponds in the abstract. Page 3: “Basing on our research about sea surface wind speed inversion model of CYGNSS sea surface data based on Machine Learning,…”: basing and based. Thank you.

Author Response

Dear reviewer,

Thank you very much for your advice. According to your suggestion, the paper has been revised. Please see the attachment. Please see the attachment.

Kind regards

Reviewer 3 Report

The proposed work aims to retrieve the wind direction taken by CYGNSS sea surface data. The work is interesting and of important application implications. I suggest accepting it after minor revision.

Additional comments:

  1. Firstly, the version of CYGNSS and reanalysis data should be introduced clearly, which is more helpful to readers. CYGNSS L1 V3.0? ECMWF/ERA5?
  2. Figure 1-4 should be added the colorbar.
  3. Figure 1-4. Are the DDM peak point and the center of mass of DDMr and DDMskirt the vertex of a rectangle? Please check again whether the marking is correct.
  4. Figure 8. The data samples are mainly concentrated in July to November. Can this phenomenon be explained from the perspective of quality control, data matching, and information extraction (φ1 and φ2)?
  5. Figure 7 is very fuzzy. please revise.
  6. Figure 10. The left parentheses of (a) should be revised.
  7. Figure 12 diff. The title needs to be expressed more clearly. According to equation (10), the diff can be calculated.
  8. Please carefully check all the reference formats!!!
  9. “calculating DDMA from the perspective of DDMA[21]. “ What is the abbreviation of DDMA?
  10. Page2, Wang should be Wang et al. the same as Pascual.
  11. “machine learning and deep learning has made it possible to build complex models.” “have”
  12. Section 3 “T(X, Y) represents Corresponds to the power”. T(X, Y) should be italicized.
  13. Page4 “namely angle φ1in Figure 1-4.” Added the space between φ1 and in. the format of φ1 and φ2 should be revised.
  14. Page5 “???? (??? , ???)and the center of mass of ???????? (???????? ,????????)to obtain”. Spaces are missed.
  15. Page7 “receiver elevation (ele) and azimuth angle (azi),φ1, φ2, leading” “Specifically, ?(?, ?)is a” Spaces are missed.
  16. Page8 “Radial Basis Function (RBF)with” Spaces are missed.
  17. Page10 “is shown in Table â…ˇ,” Should be Table 2.
  18. Page 11 “It can be seen from the figure that” Should be Figure 10(a)
  19. Table3 or table3 should be Table 3.

Author Response

(The authors gave the same response as above.)

Round 2

Reviewer 1 Report

The authors addressed the main concerns from the reviews, and the revised version looks good. I recommend it to be published as it is.

Reviewer 2 Report

The authors have answered proficiently to all points raised. I have no further comments. In my opinion, the paper is ready now to be published after minor corrections to grammatical errors.